
# The regional impact of urban emissions on climate over central Europe: present and future emission perspective

Peter Huszár[1], Michal Belda[1], Jan Karlický[1], Petr Pišoft[1], and Tomáš Halenka[1]

[1]Department of Atmospheric Physics, Faculty of Mathematics and Physics, Charles University, Prague, V Holešovičkách 2, 180 00 Prague 8, Czech Republic

*Correspondence to:* P. Huszar (peter.huszar@mff.cuni.cz)

**Abstract.** The regional climate model RegCM4.2 was coupled to the chemistry transport model CAMx, including two-way interactions, to evaluate the regional impact of urban emission from Central European cities on climate for present (2001-2010) and future (2046-2055). Short-lived non-$CO_2$ emissions are considered, and, for the future impact, only the emission changes are accounted for (the climate is kept 'fixed'). The urban impact on climate is calculated with the annihilation approach, when two experiments are performed: one with all emissions included and one without considering urban emissions. The radiative impacts of non-$CO_2$ primary and secondary formed pollutants are considered: namely ozone ($O_3$), sulfates (PSO4), nitrates (PNO3), primary organic and elementary carbon (POA and PEC).

The validation of the modeling system is limited to key climate parameters, near surface and precipitation. It shows that the model, in general, under-estimates temperature and overestimates precipitation. We attribute this behavior to too much cloudiness/water vapor present in the model atmosphere as a consequence of over-predicted evaporation from the surface.

The impact on climate is characterized by a statistically significant cooling up to -0.02 K and -0.04 K in winter (DJF) and summer (JJA) season, mainly over cities. We found that the main contributor to the cooling is the aerosols direct and indirect effects, while the ozone titration, calculated especially for DJF, plays rather a minor role. In accordance with the vertical extent of the urban emission induced aerosol perturbation, cooling dominates the first few model layers up to about 150 m in DJF and 1000 m in JJA. We found a clear diurnal cycle of the radiative impacts with maximum cooling just after noon (JJA) or later in afternoon (DJF). Furthermore, statistically significant decreases of surface radiation are modeled, in accordance with the temperature decrease. The impact on the boundary later height is small but statistically significant and reaches -1 m and -6 m decreases in DJF and JJA, respectively. We did not find any statistically significant impact on precipitation and wind speed. Considering future emissions, the impacts are, in general, smaller - as a consequence of smaller emissions resulting in smaller urban induced chemical perturbations.

In overall, the study suggest that the non-$CO_2$ emissions play rather a minor role in modulating regional climate over central Europe. Much more important is the direct climate impact of urban surfaces trough urban canopy meteorological effects as we showed earlier.





# 1 Introduction

High population densities in urban areas and, thus, concentrated human activities result in an intense emission source that cities represent. United Nations reports year 2009 as the first with more the 50% of the Earth population living in cities (UN, 2009). Therefore, understanding their impact on the environment, both within the city itself and on larger scales outside of it,

is gaining huge importance.

Among the many types of urban impact on environment, Folberth et al. (2015) defines the impact on the atmospheric environment as the 'most important and most far-reaching'. Cities influence both the meteorology and atmospheric chemistry in many ways: (i) urban areas are largely covered by artificial surfaces, being clearly distinguished from natural ones by mechanical, radiative, thermal, and hydraulic properties. These surfaces affects the mechanical and thermodynamical properties

of the air atmosphere above in a very specific way (Lee et al., 2011; Huszar et al., 2014). (ii) Cities further emit large amount of gaseous material and aerosol into the air directly influencing air-quality and atmospheric chemistry in general (Timothy and Lawrence, 2009; Huszar et al., 2016). (iii) At last, direct emissions of greenhouse gases (GHG) and aerosol, and the urban emission induced perturbation of secondary formed radiatively active gases and aerosol lead to modification of radiative and thermal balance, cloud properties and climate (Seinfeld and Pandis, 1998; Folberth et al., 2015).

The complex nature of the influence of the urban-canopy on meteorological conditions and climate has been identified since the early 80s along with the description of the urban heat island effect (UHI; Oke, 1982). Since than, many studies examined UHI and related urban impacts on meteorology: e.g. on temperature (Basara et al., 2008; Gaffin et al., 2008); humidity (Richards, 2004); turbulence (Roth, 2000; Kastner-Klein et al., 2001); the overall structure of the boundary layer (Angevine et al., 2003); wind speed (Hou et al., 2013); precipitation and hydrological cycle (Rozoff et al., 2003). Along with

the development of numerical methods, and, especially with the introduction of simple-to-complex urban-canopy parameterizations/models, modeling approaches to urban effects on meteorology became widespread examining local (Flagg and Taylor, 2011; Wouters et al., 2013; Hou et al., 2013) and regionals scales as well (Feng et al., 2013; Trusilova et al., 2008; Struzewska and Kaminski, 2012; Huszar et al., 2014). The listed studies (and many more – see references within) show that the cities impact on the meteorology and climate is usually not limited to the geographical location of the city itself, but the impact

propagates to regional scales and that urbanization can contribute to regional warming (e.g. Huszar et al., 2014).

Regarding the second path-way (ii), urban emissions impact the composition of the air and chemistry, in general, on every scales from local to global, depending on location of the city and on the lifetime of the emitted or secondary produced pollutants (Baklanov et al., 2010; Folberth et al., 2015). Long-lived species emitted by large cities like carbon-dioxide ($CO_2$) affect the chemical composition over whole globe (e.g. Lawrence et al., 2007; Folberth et al., 2012) while the urban impact concerning

short lived pollutants like nitrogen oxides (NOx), ozone ($O_3$) is more limited to local and regional scales (Im and Kanakidou, 2012; Im et al., 2011a, b; Huszar et al., 2016).

There is a large number of studies that focused on the impact of short lived pollutants from cities on local, regional and even global scale. Measurement based studies investigated the differences between the urban plume and the background environment composition (e.g. Freney et al., 2014; Molina et al., 2010; Kuhn et al., 2010). There has been also model based efforts to





estimate the cities fingerprint on the atmospheric chemistry across multiple scales: on global scale, Lawrence et al. (2007), Butler and Lawrence (2009),Folberth et al. (2010), Butler et al. (2012) and Stock et al. (2013) gave estimates on the city emissions impact on the environment. On regional scales, many studies focused on European urban centers (e.g. Hodneborg et al., 2011; Im and Kanakidou, 2012; Finardi et al., 2014; Skyllakou et al., 2014; Markakis et al., 2015; Markaris et al., 2016), on megacities in eastern Asia (Guttikunda et al., 2003; Tie et al., 2013) and Mexico city (Li et al., 2011).

Much less work has been done regarding the climate impact of urban emissions and consequent chemical perturbations (pathway iii). Recently, two reviews were published addressing the urban emission impact on climate (among other impacts): Folberth et al. (2015) and Baklanov et al. (2016). Focusing on so called megacities (inhabitants larger than 10 million), both agree, that there is high confidence on the climate impact of $CO_2$ emitted and that much of the radiative impact of urban emission is attributable to this gas. Folberth et al. (2012) estimates the global radiative forcing (RF) to 120.0, 28.4 and 3.3 $mW\,m^{-2}$, respectively, from the long-lived components $CO_2$, methane ($CH_4$) and nitrous oxide ($N_2O$) emitted by megacities. For the impact of short-lived species, namely ozone and aerosol, they give a global mean RF 5.7±0.02 $mW\,m^{-2}$ due to the increase in tropospheric ozone, and -6.1±0.21 $mW\,m^{-2}$ due to urban induced aerosol increase. Comparing these last two numbers, it is clear that on global average, ozone and aerosol climate effects nearly cancel each other.

On local and regional scale, however, the impact of short lived pollutants can be substantially more important (Baklanov et al., 2016). As a result of NOx and non-methane volatile organic compound (NMVOC) emissions, ozone forms and under favorable weather conditions it can accumulate over and around cities. Consequently, it can trigger positive radiative forcing that results in local/regional warming (Park et al., 2001). Many studies showed significant effect of aerosol originating from cities or highly populated agglomerations on radiation and consequently meteorology and climate. Black carbon from an Indian megacity was investigated by Tripathi et al. (2005). Giorgi et al. (2002) attributed observed cooling in a highly populated area in China to sulfate and carbonaceous aerosol originating from the same area. Ramanachan and Kedia (2010) looked at black carbon pollution over Ahmedabad, India and indicated its important role in the radiative budged possibly affecting monsoon rainfall. Roldin et al. (2011) calculated the radiative impact of aging urban plumes from Malmö, Sweden and found a RF from -0.3 to -3.3 $mW\,m^{-2}$ depending on the distance from city and the specific cloud properties.

As already said, the globally averaged impact of short-lived gases/aerosols is of rather minor importance. However, previously listed studies prove that their effect can be significant locally, therefore it is important to examine the distribution of the climate impact of urban emission on smaller scales (than global) as well. Furthermore, these studies calculated the individual impact of selected pollutants, but, given the non-linear nature of the emissions-on-climate impact, the overall impact may somewhat differ from that one calculated as the sum of individual impacts from ozone and aerosols. Many studies looked at the chemical perturbations introduced by urban emissions, as already listed before. However, none of them was interested in the radiative impacts of these perturbations, or more specifically, how the ozone (which can include both decreases and increases) and aerosol concentration changes affect the radiative balance and consequently the climate. We introduce this study as one of the firsts that quantifies the climate impact of urban emission on a regional scale. We use an interactively coupled regional climate-chemistry modeling framework. Indeed, given the strong tights between emission, air-chemistry and radiative impacts, an integrated approach is required. Our focus of region is central Europe with middle sized cities often exceeding 1 million



inhabitants. The paper is a follow-up study to Huszar et al. (2016) where we were interested on the air-quality changes induced by urban emissions. In this study, as a further step in analyzing the results, we investigate the consequent climate impacts. Besides the present day urban impact, we will also examine, how will this impact change considering future emissions of short-lived (non-$CO_2$) pollutants.

## 2 Models and experimental design

### 2.1 Models

Models and their configuration are identical as in Huszar et al. (2016). Here we give, thus, a rather brief description only and for more detailed insight, refer to that study. As meteorological driver, regional climate model RegCM4.2 was applied (Giorgi et al., 2012) with the following parameterizations: surface processes - BATS scheme Dickinson et al. (1993), large-scale precipitation – SUBBEX (Pal et al., 2000), convection – Grell scheme (Grell, 1993), planetary boundary layer – Holtslag scheme (Holtslag et al., 1990). The radiative transfer is calculated with the NCAR Community Climate Model Version 3 (CCM3; Kiehl et al., 1996). This scheme describes the effect of different GHG, cloud water and cloud ice. Cloud radiative properties are calculated according to cloud liquid water content and effective droplet radius, while the fraction of cloud ice is diagnosed as a function of temperature. The radiative properties of sulfate aerosol are calculated according to Kiehl et al. (2000). For carbonaceous aerosols, optical properties were extracted from the biomass-burning study of Reid et al. (2005). By default, climatological zonally averaged ozone profiles are prescribed in RegCM after Dütsch (1978). We further did not consider urban canopy related meteorological effects (as interested purely on the effect of emissions), i.e. the SLUCM urban model implemented under BATS (Huszar et al., 2014) was not invoked.

The model RegCM4.2 has been online coupled to chemistry-transport model (CTM) CAMx. CAMx is an Eulerian photochemical CTM developed by ENVIRON Int. Corp. (http://www.camx.com). CAMx includes multiple gas phase chemistry mechanism options (CB-IV, CBV, CBVI, SAPRC99). In this study the CBV scheme (Yarwood et al., 2005) was used. CAMx further implements multi-sectional or static two mode particle size treatments, wet deposition of gases and particles, plume-in-grid (PiG) module for sub-grid treatment of selected point sources, Ozone and Particulate Source Apportionment Technology, mass conservative and consistent transport numerics as well as parallel processing. To calculate the composition and phase state of the ammonia-sulfate-nitrate-chloride-sodium-water inorganic aerosol system in equilibrium with gas phase precursors, the ISORROPIA thermodynamic equilibrium model (Nenes and Pandis, 1998) is implemented.

The two models are coupled using the technique of online access coupling defined by Baklanov (2010). Data between models are exchanged on a hourly basis: RegCM4.2 is ran for one hour, after CAMx is driven with the output of the RegCM4.2 run. The calculated chemical concentrations are, after an hour, supplied back to the RegCM4.2's radiation code. The whole cycle is then repeated. According to Grell and Baklanov (2011), this is a justifiable update time for the resolution used. The species provided by CAMx to RegCM4.2 and, thus, considered in radiation calculations in CCM3 are: ozone, secondary inorganic aerosols – sulfates (PSO4) and nitrates (PNO3), and, primary organic and inorganic carbon (OC and BC). In the used climate model, the radiative effects of only PSO4, BC and OC are included, while for nitrates we slightly modified the sulfate optical





properties following McMeeking et al. (2005). Further, to include the indirect effects of secondary inorganic aerosols, we implemented the work of Giorgi et al. (2002). They considered only sulfates, however, we used their approach for nitrates as well with a slight modification following Wang et al. (2010). We denote this coupled system as RegCMCAMx4 and it is basically an advanced version of the original RegCMCAMx introduced by Huszar et al. (2012). In further text, the term

'model' will always refer to the RegCMCAMx4 coupled modeling system.

## 2.2 Experimental set-up

The model was run at 10 km x 10 km horizontal resolution centered over Prague, Czech Repubic with 160x120x24 (in x , y , and z direction) gridboxes. The uppermost level for the climate is at 50 hPa, while the chemistry was integrated only on the lowermost 16 levels (approximately up to 300 hPa). Numerous experiments were carried out for the 2001-2010 and a future,

2046-2055 period. As meteorological initial and boundary conditions (ICBC) we used the ERA Interim reanalysis (Simmons et al., 2010). For chemical ICBC, we performed a larger, 30 km x 30 km, domain run covering the whole Europe. This choice was motivated by acquiring better ICBC for chemistry if the chemical processes are developed on a much larger domain. This large domain run was driven by time-space invariant chemical ICBC, which resulted in some model biases detailed by Huszar et al. (2016), especially regarding ozone. Biogenic emission of isoprene and monoterpenes were computed online according to

Guenther et al. (1993).

As we were interested in how the urban impact will change in future due to modified emissions, we used the same meteorological and chemical ICBC for the future 2046-2055 period. Only the anthropogenic emissions used were different (see further). This choice was further based on the assumption that the (mainly) long-lived GHG driven climate change will not cause significant differences in climate between years 2046-2055 vs. 2001-2010 and that the added change triggered by urban

emissions will depend mostly on the emissions and not on the background climate.

## 2.3 Emissions

The TNO emissions prepared in the framework of the FP 7 MEGAPOLI project (Kuenen et al., 2010) were used in this study. This high resolution ($1/8°$ longitude $\times$ $1/16°$ latitude, roughly $7\,\mathrm{km} \times 7\,\mathrm{km}$) European emission database provides annual emissions estimates for $NO_x$, sulfur dioxide ($SO_2$), NMVOC, $CH_4$, ammonia ($NH_3$), carbon monoxide (CO) and $PM_{10}$ and

$PM_{2.5}$ in 10 activity sectors. Additionally, future sector- and country based scaling factors are introduced for reference years 2020, 2030 and 2050. For the energy sectors, the scenarios have been generated based on energy model runs using the Pan-European TIMES energy system model (Blesl et al., 2010). A moderate climate policy is assumed, which gives options of applying further mitigation strategies. In case of non-energy related sectors, scaling factors were calculated with the GAINS model (Amann et al., 2008) or other assumptions. A detailed description of the procedure, how emission factors and future

scaling factors were extracted is provided by Theloke et al. (2010).

TNO annual emission data were regridded into the model grid. After, sector specific temporal disaggregation factors and NMVOC speciation profiles were applied to decompose the annual sums into hourly emissions according to Winiwarter and Zueger (1996). Fig. 1 presents the annual emission interpolated to our grid for gaseous pollutants and PM2.5 for the base 2005



year and the difference between 2050 and 2005. It clearly indicates emission reductions for every species. However, due to increased energy demand in future, some sectors like sector 2 – small and medium combustion plants, sector 3 – industrial combustion plants as well as sector 7.3 – LPG road transportation, or sector 8 – non-road transport (incl. airport traffic) and mobile machinery, will produce more emissions. This is clear from NMVOC, NOx, PM2.5 as well as $SO_2$, especially for
southeastern countries in Europe.

## 2.4  Experiments

Several experiments were performed for the two mentioned periods (2001-2010 and 2046-2055), i.e. around the reference emission years 2005 and 2050. Adopting the 'annihilation approach' (Baklanov et al., 2016), first a run with all emission was carried out for both periods, denoted 05BASE and 50BASE (referring to the last two digits of the emission years). After,
by removing urban emission for large cities (we used the threshold 500 000 inhabitants for western and 100 000 for eastern Europe; for more details see Huszar et al. (2016)), 'zero' experiments were performed for both periods: 05ZERO and 50ZERO.

In order to identify the climate signal of urban emission induced perturbations, one have to account for the effect of the variability of the modeled climate. This can be achieved by performing ensembles of a few members for all experiment. The goal is to separate the signal information from the noise. However, there is no 'a priori' knowledge of how many members are
required so we decided, on the basis of our computer and time sources, to run ensembles with 3 members, similarly as in Olivie et al. (2012) or Huszar et al. (2013) who looked at transport emissions impact on climate. We have chosen the ensemble mode rather than adopting the scaling approach which is based on the assumption that the forcing (emission in this case) and signal are linearly connected.

The climate is usually defined as 'average weather' (including extremes) over 30 year. We selected only a 10 yr long period
which represents the current conditions in our study. It has to be thus justified that this decade is characteristic within a longer period (i.e. whether it is not too warm, or extraordinarily cold etc.). Furthermore, in future period, only emission changes are considered, so we are also interested in the possible modification of the climate over the region in focus. Fig. 2 shows the 1970-2050 annual and 10 yr running mean of near surface temperature averaged over our domain calculated from the EURO CORDEX regional climate models ensemble consisting of 25 members (RCM-GCM combinations) following the RCP 4.5
scenario (Jacob et al., 2013; Kotlarski et al., 2014). According to this figure, the 2001-2010 average temperature (about 8.5 °C ) is characteristic or typical for the respective longer 30 yr time periods. Further, the increase of temperature for the 2046-2055 period is about 1 °C. A similar temperature increase was modeled for a very similar domain (and at the same resolution) in Juda-Rezler et al. (2012) for the 2040-2050 period. They investigated the impact of future climate change on air-quality and found that for the mentioned period this impact is insignificant.





## 3 Results

### 3.1 Model validation

We limit the model validation to the meteorological model output only. A detailed validation of the 'chemical' results of the presented experiments was performed in Huszar et al. (2016).

5     The average 2001-2010 model results are compared to E-OBS gridded observational data: seasonal differences for the near surfaces temperature and total precipitation are presented in Figures 3 and 4, in °C and $\mathrm{mmday}^{-1}$, respectively. Temperature is mostly under-predicted by around -0.5 to -1.5 °C, mostly in DJF and MAM (spring) and in a slightly smaller extent in JJA and SON (autumn). What is however more striking is the large negative, but also positive biased behavior over the Alps where the model-observation difference can reach 4 °C in absolute sense.

10     Precipitation is usually over-predicted over the domain, mostly during MAM and JJA, by up to 2-3 $\mathrm{mmday}^{-1}$. Again, larges biases are modeled over the mountainous regions, mainly Alps but also over the eastern Carpathians at the southeastern edge of the domain, where it can reach 4 $\mathrm{mmday}^{-1}$ in absolute numbers (precipitation decreases are modeled as well).

    The negative temperature bias is well seen on the annual temperature cycle in Fig. 5 (upper). The domain averaged bias becomes almost zero during months May-July and is largest during the preceding (February-April) and following months 15 (August). The over-prediction of precipitation is striking on the annual cycle plot. During the warm season, it can reach 60-80 %. The encountered model biases including their causes are discussed later.

### 3.2 The modeled species concentrations

As the climate impact of urban emissions is calculated considering the radiative effects of $O_3$, sulfates (PSO4), nitrates (PNO3), primary organic aerosol (POA) and primary elementary carbon (PEC), we will show the distribution of these species only. The 20 horizontal distribution of the average DJF and JJA impact on surface concentrations for the present and future period is plotted in Fig 6 and 7. In winter, ozone is strongly titrated by urban emissions decreasing it by up to 10 ppbv over urban centers, corresponding to -50 % decrease. However, ozone decreases over rural areas as well often by 5-10 %. In JJA, ozone titration dominates over urban centres (with up to 50 % decrease similar to DJF), however, further away over rural regions and especially over the eastern part of the domain, a slight ozone production occurs up to 0.5 ppbv corresponding to 0.5-1 % increase.

25     The city emission's contribution to PSO4 burden in DJF reaches 2-3 $\mathrm{\mu g\,m}^{-3}$ over cities in western Germany, and is usually 0.4 $\mathrm{\mu g\,m}^{-3}$ over other cities, contributing to the total sulfates by up to 30 %. The JJA sulfate enhancement due to city emission is larger than in DJF, reaching 4 $\mathrm{\mu g\,m}^{-3}$ (or 40 % contribution) over urban areas.

    Urban emission contribute to nitrates in DJF usually by more than 5 %, while in the cities this increases up to 70 %. In absolute numbers, this means about 1-1.5 $\mathrm{\mu g\,m}^{-3}$ increase (most dominantly over the Ruhr area in Germany). In JJA, 30 absolute contribution of urban emission to the total PNO3 is larger and exceeds 0.2 $\mathrm{\mu g\,m}^{-3}$ over rural areas as well. However, in relative numbers, the contribution is smaller than in DJF, with maximum values over eastern Europe (up to 20 %).

    The urban emissions contribution to POA and PEC is similar, being slightly higher in winter months (4-6 % over large areas). Over cities, the contribution reaches 2 $\mathrm{\mu g\,m}^{-3}$ for both species and seasons, corresponding to 70 % contribution.



The vertical distribution of species averaged over the domain y axis (i.e. in S-N direction) is plotted in Figures 8 and 9 for DJF and JJA, respectively. The saturated NOx conditions causing ozone titration are characteristic for the lower model levels, especially for the DJF month causing $O_3$ decreases reaching -0.8 ppbv up to 150-200 m in vertical, corresponding to 5% decrease due to urban emissions. Further, in winter, the whole modeled air over the domain is subject to ozone destruction. In JJA, $O_3$ destruction over urban areas takes place mainly over the western part of the domain (up to 300 m in height). Over the eastern part, ozone is usually increases on all model levels with most ozone produced (up to 0.2 ppbv) between heights 100-1000 m. This represents and increase up to 0.5 %.

The urban induced sulfate contribution to the total PSO4 is limited to lower heights in DJF (1 % contribution up to 1 km) than in JJA (1 % contribution up to 7 km). However, in absolute numbers, the vertical contribution is only slightly higher in JJA than in DJF, and usually negligible over 1 km. The absolute nitrates contribution (3rd column in both figures) is very low in DJF and is much significant in JJA. However, in relative numbers, due to lower background PNO3, the winter urban contribution remains larger than 8 % up to 1.7 km in vertical. Urban emissions contribute to black and organic carbon in a similar manner with significant contribution in the lowest 2 model levels (i.e. up to 150 m) and the contributions are higher in DJF. The PEC and POA contributions are higher than 1 % for the whole vertical extent of the domain in JJA, while in winter this is true up to only about 5 km.

According to Fig. 1, future NOx and NMVOC emissions usually decrease, except for eastern Europe. This is reflected in the urban impact on ozone in the future period. In winter, the modeled ozone titration is smaller in magnitude for the whole domain, as seen on Fig. 6 as well as on the vertical distribution in Fig. 8. According to Fig. 7 and 9, the future urban emission induced ozone changes in JJA are, similarly to DJF, smaller in magnitude. This means reduced titration over cities, however the slight ozone production seen especially over eastern Europe is suppressed as well.

Looking at sulfates in both seasons, due to drastic decrease of $SO_2$ emissions, a clear decrease if modeled for PSO4 as well, especially over eastern Europe, were 2005 $SO_2$ were still high and emissions control measures had a larger effect. Sulfates increase only over a very few urban areas, where $SO_2$ emission are mainly from SNAP activity sectors 2 and 3 that increase emissions in future, as projected by TNO (see Sec. 2.3).

A somewhat different change is projected for the urban induced PNO3 change. While in summer, a decrease is simulated due to NOx emissions reductions, the picture is more complicated in DJF. Nitrates are formed via neutralization of nitric acid by available ammonia forming ammonium-nitrate $NH_4NO_3$ (Wang et al., 2013). As in future less $SO_2$ are projected in general, less ammonia will neutralize sulfates (forming ammonium-sulfates, $(NH4)_2SO_4$), thus more is preserved to form ammonium nitrates. This might explain the winter increase of nitrates in future due to urban emissions, despite the fact that NOx emission are reduced. This increase is most evident over eastern Europe, where the sulfate decrease is the largest and ammonia emission will increase slightly.

The urban induced black and organic carbon perturbation for future emissions in DJF shows a slight decrease over easter Europe due to generally larger emissions of PM2.5, of which PEC and POA represents an important fraction. In summer, the 2046-2055 PEC and POA urban perturbations are rather similar to present day values with larges decreases over wester Europe.





### 3.3 Impact on climate

By absorbing the long-wave radiation from the surface, ozone acts as a greenhouse gas. Aerosols considered are interacting with short-wave solar radiation directly or by modifying the clouds optical properties. Consequently, their perturbation shown in the previous subsections, can lead to perturbation in the radiative balance of the atmosphere and hence may impact climate.

Here we will show the impact for the cold and warm periods of the year (DJF and JJA) on following parameters: near surface temperature, temperature on higher model levels and planetary boundary layer height (PBL). The impact on precipitation and wind speed is not shown as they showed to be very noisy and statistically insignificant in both DJF and JJA for the present, as well as for the future period. In each figure, shaded areas represent statistically significant changes on the 98 % confidence level.

Fig. 10 shows the DJF and JJA impact of urban emissions on the near surface temperature for the 2001-2010 and 2046-2055 periods. In winter, a clear temperature decrease is modeled which is statistically significant especially over eastern Europe (mainly Slovakia and Hungary), reaching -0.015 K change. A relatively stronger cooling is calculated also over the Ruhr area in Germany, up to -0.02 K. Larger continuous areas exhibit cooling around -0.005 K. The JJA picture about the urban emission's climate impact is more complicated. Although cooling still dominates the impact, reaching -0.04 K, especially over south eastern part of the domain, the area of significance is more fragmented and sometimes a warming is modeled as well. A few cities are identifiable where cooling reaches local maximum: e.g. Warsaw, Berlin, Katowice, Prague. With future 2050 emissions, the impact on temperature is generally smaller but keeping the pattern from the present-day period: i.e. strongest cooling over eastern Europe reaching -0.01 K. The future JJA impact is, again, resembling the present day impact in terms of the spatial pattern, with strongest statistically significant cooling over eastern/southeastern Europe. It reaches -0.04 K, similarly as in the present day, but the corresponding areas with such cooling are clearly smaller than in the present day period.

To illustrate the importance to perform ensemble simulations and calculate the signal from ensemble mean in order to extract the climate signal from the noise, we present in Fig. 11 the 2001-2010 JJA near surface temperature impact from individual ensemble simulations which means 3 × 3 possible outcomes. The figure reveals substantial differences at some areas between individual possibilities of calculating the temperature change. While in each plot, cooling dominates and over some areas (like the Balkans) it is significant in each one, its extent and distribution differs largely. There are also differences in the areas of warming: over northern Italy, it can reach sometimes 0.08 K. The prevailing cooling also reaches higher absolute values (up to 0.06–0.08 K) in individual realizations than calculated from the ensemble means (Fig. 10).

The vertical cross-section of the impact averaged over the domain's y-axis is plotted in Fig. 12. The temperature decrease due to urban emissions is significant only on the lower levels, up to 200 m in DJF and 1000 m in JJA. It is stronger over the eastern part of the domain. Comparing to the vertical distribution of the emission impact on concentrations (Fig. 8 and 9), the statistically significant decrease corresponds to the vertical extent of the urban emission induced species perturbation, especially for secondary inorganic aerosols. For future emissions, the impact is of smaller magnitude and statistically significant only below the first two model levels in DJF (up to 150 m). In JJA, the significant impact spreads to higher levels but is limited mostly to the eastern part of the domain. Note that only the first 13 RegCM model levels are plotted.





The impact on surface short-wave (SW) radiation is plotted in Fig. 13. Due to urban induced aerosol increase, decrease in surface solar radiation is computed in both DJF and JJA. Especially in DJF, the hot-spots of SW decrease well correspond to the location of large cities (up to $0.6\,\mathrm{Wm^{-2}}$ decrease). In summer, decrease of SW is significant mainly over cities but is larger in magnitude than in DJF, reaching $-1.8\,\mathrm{Wm^{-2}}$ over the Ruhr area (Germany), Warsaw (Poland) and is high above Berlin (Germany) as well, reaching $-1\,\mathrm{Wm^{-2}}$. The surface short-wave radiation decrease due to future urban emissions is smaller than the present day impact (by about 50 % in JJA), which is in line with the modeled temperature decrease. Only over parts of central and over eastern Europe, SW radiation decreases due to cities more than in present-day, which is in good correspondence with the increased particulate emissions (seen in Fig.1) and consequently increased urban induced perturbation.

The impact on the height of the planetary boundary layer (PBL) is shown in Fig. 14. It is characterized by a small but statistically significant decrease in DJF especially over urban areas up to -2 m. In JJA, the decrease dominates and reaches -6 m (over the Balkan region, Ruhr area, Berlin, Warsaw etc.). Certain areas exhibit increases of PBL height (over western Poland, northern Italy) which is probably due to non-linear feedbacks. These PBL increases, however, cover only a very limited area. The impact PBL height in future DJF reaches similar or slightly smaller values, which is also visible for JJA, when PBL decreases significantly over a fragmented regions across the domain, with a slight dominance over easter Europe.

We are further interested in whether the temperature impacts seen in Fig. 10 and 12 exhibit some diurnal variation. Thus, we plotted in Fig. 15 the average diurnal cycle of the vertical profile of the domain averaged impact on temperature for DJF and JJA seasons (only for the present day period). The simulated cooling seen in Fig. 12 is captured in diurnal cycle as well, but most importantly, this figure indicates that the cooling is most dominant during afternoon hours with maximum values 14:00-15:00 UTC (3-4 pm local time) in DJF and a bit earlier, at 13:00-14:00 UTC (2-3 pm. local time) in JJA. Another interesting feature is a forming of warmer air above the cooling layers, which develops almost along with the maximum cooling during afternoon hours: in DJF these warming reach maximum slightly shifted to earlier times, while in JJA, it occurs a bit later than the cooling maximum. The layer of warming is thicker in DJF (around 500 m) while in JJA it reaches around 100-200 m thickness. A further feature is a slight warming on the top of the domain in DJF, however it occurs on the last model layer and is most probably an artifact in connection with the boundary conditions imposed.

To gain an idea, how the urban induced climate changes (in connection with the chemistry perturbations seen in Fig. 6-9) contribute to the radiative effects of all aerosols, we plotted the near surface temperature impact of all sulfates, nitrates, black and organic carbon, i.e. not only those induced by urban emissions (Fig. 16). Caused by these aerosols, statistically significant cooling dominates the whole domain in both seasons. In DJF, it reaches cooling over 0.2 K over most of the modeled region, being the highest over eastern Europe (up to -0.4 K cooling). In JJA, the cooling is of smaller magnitude (usually over 0.1 K), being highest over central and southeastern Europe (up to -0.2 K).

## 4 Discussion and conclusions

The evaluation of the RegCMCAMx modeling system from the meteorological point of view showed two main bias behaviors: a general underestimation of near surface temperatures and a large overestimation of precipitation. Over Europe, Torma et al.





(2008) and later Torma et al. (2011) modeled the present day climate over central Europe with an earlier version of RegCM at the same horizontal resolution and found this negative bias as well. They showed that this underestimation was a result of over-estimation of cloudiness causing less solar radiation insolating the surface. The enhanced cloudiness is linked also to the large positive precipitation bias both Torma et al. (2011) and us modeled. They tried to reduce it by tuning parameters controlling

cloud/precipitation microphysics; in particular, reducing the autoconversion rate, increasing the raindrop evaporation rate and reducing the raindrop accretion rate. However, after this tuning some positive precipitation still remained. The overpredicted cloudiness and resulting precipitation is probably a consequence of too much water vapor entering the air from the surface. Indeed, Winter et al. (2009) showed that the BATS surface scheme (used here and also by Torma et al.) largely overestimates evapotranspiration and the associated latent heat flux (which also contributes to the negative surface temperature bias). Later,

Wang et al. (2015) confirmed this as well by comparing the newer CLM4.5 land-surface module with the older BATS within the RegCM4 model. Our model further encounters large biases over mountainous areas, mainly the Alps. It has to be realized that the model orography here, even at 10 km x 10 km resolution, is relatively smooth and not able to resolve mountains peaks and valleys correctly, hence the model surface is either much lower or much higher than the real orography, which determines significantly the observed temperature (in EOBS data).

The modeled ozone distribution showed that urban emission over central Europe cause, in average, ozone decrease due to titration. Im et al. (2011a, b) came with a similar conclusion: due to reaction with NO, ozone is destroyed near the sources, however where NOx is further mixed with additional NMVOC (including of biogenic origin), the odd oxygen present in $NO_2$ becomes available to produce ozone further downwind. The vertical figures reveal that this happens mainly over eastern Europe with less dense urban emissions and with more rural air available, and, further at higher altitudes.

The aerosols perturbation for the primary aerosols (black and organic carbon) is limited to the source regions and spans a small vertical extent, mostly residing in the boundary layer that is, in general, tinner in DJF than in JJA. The secondary formed sulfates and nitrates of urban origin occupy much ticker atmospheric layer especially in JJA, due to higher vertical mixing in summer. Sulfate concentrations differ between the two seasons only slightly, as modeled similarly by Schaap et al. (2004). On the other hand, the nitrates show a significant seasonal variation. The low values in winter are connected to

much less available ammonia emissions that would form ammonium-nitrates. Elevated ammonia concentrations are essential to stabilize this aerosol at high temperatures, thus higher concentrations of PNO3 are modeled especially over western Europe were ammonia emission are also high in JJA (Schaap et al., 2004).

The radiative and, consequently, temperature response is well explained by the urban induced chemical perturbations. Ozone is mostly destroyed over lower model levels and the production over higher altitudes is very small. This may result in an

overall negative radiative forcing and thus cooling. However, long-wave heating rates are low above the surface due to similar temperature of air and the surface (Petty, 2006). Thus the modeled ozone perturbations are expected to have a very small radiative impact.

Aerosol perturbation are purely positive and are limited to the few model levels above the surface. Due to direct and indirect radiative effects and due to the fact that surface albedos over the examined regions do not have high reflectivity (Garcia et al.,

2012), this has to result in decrease in solar radiation impacting the surface and hence in reduced temperature. The temperature



response confirms this. Although the impact is very small in DJF (of order of $10^{-3} \pm 10^{-2}$) but statistically significant almost over the whole domain. In winter weather is characterized with more stable conditions and reduced variability, thus the climate signal of the urban enhanced aerosol is distinguishable from the reference climate, even having a very small magnitude.

For major urban emitters like Warsaw, Berlin, or the Ruhr area, the temperature response in winter resembles the location of significant solar short wave radiation decrease. In summer however, although the SW radiation at surface decrease significantly over many urban areas, the resulting surface temperature response has a different spatial distribution and its maximum do not follow the location of negative peaks in the SW figure. Zanis (2009) and Zanis et al. (2012), who examined the direct radiative impact of anthropogenic aerosol over Europe, came with a similar finding: i.e. the geographic pattern of the aerosol optical depth (AOD) and the changes in the top-of-the-atmosphere radiative forcing (TOA RF) do not correspond to he near surface temperature changes and they explain it by the complex role of aerosol in modifying not only the radiative budget, but having dynamical feedback on the whole atmospheric circulation. Similarly, Roeckner et al. (1999) found a poor correlation between the distribution of forcing and the temperature response in their simulations. Recently, Forkel et al. (2015), using the WRF model, examined the direct/indirect aerosol feedbacks on European meteorological conditions and arrived to similar conclusion regarding the aerosol distribution and the resulting radiative and climate effects.

The simulated changes in the boundary layer height follow from the overall decrease of lower tropospheric temperatures. The scattering and absorption of aerosols reduces surface radiation, which in turn suppresses sensible heat flux (Yu et al., 2002). The reduced sensible heat reduces temperatures and limits the growth of the boundary layer. The decrease is highest (in both seasons) over urban areas and is much higher in summer when temperature decrease is also higher.

The vertical extent of the statistically significant radiative changes (cooling) is limited to a few model levels in the lower troposphere. This corresponds to the vertical distribution of the chemical perturbations triggered by urban emissions, which encompasses a thicker layer in JJA than in DJF – similarly, the negative temperature response reaches higher levels in summer, accordingly. Although it turned to be statistically insignificant on the 98% confidence level, a slight warming is modeled over the region mentioned previously. This is not visible on the vertical cross-section figures (due to significance shading), however, it is visible on the diurnal cycle plot where the significance test was omitted. Earlier, Nguimdo and Njomo (2013) found also strong cooling for the lower atmosphere, with slight heating at the upper atmosphere. The cooling in aerosol layer is caused by the reflection of solar radiation from the aerosols, which reduces the amount of solar radiation available for absorption. Above the aerosol layer (in our case the urban triggered enhanced aerosol layer), the reflected radiation becomes available for absorption thus leading to slight heating in the corresponding atmospheric layers.

We found that the urban emission induced cooling in the lower troposphere has a clear diurnal cycle. It follows the variation of the incident solar radiation during the day. In winter, the maximum cooling is shifted towards later hours which can be explained by the delayed propagation of aerosol signal through the boundary layer that is often characterized by inversion in this season. Unlike in winter, in summer the cooling occurs almost simultaneously with the maximum insolation values.

The future impact is, in terms of horizontal and vertical distribution, very similar to the present day impact. However, due to lower emission in general, it is smaller in magnitude. This is true especially for eastern European countries, where current emissions are still high and rapid reductions are expected in the future (mainly for $SO_2$, as the main precursor for sulfates).



The radiative impact of aerosols triggered by all emissions on surface temperatures is very similar to numbers in Zanis et al. (2012) or Forkel et al. (2015). Comparing to the urban emission impact, our results indicate that urban aerosol (including the secondary ones) account for about 10 % of the total cooling. The question is, why urban aerosols cause larger cooling during JJA than in DJF, in opposite to the cooling caused by all the aerosols which is stronger in winter than in summer. This can be

explained by the different relative contribution of urban particulate matter to the total aerosol burden. It is clearly seen on the vertical cross-section of the perturbation that in JJA, the relative contribution of the urban aerosols is, in general, higher at each model levels than in DJF (please note the different color scales). This is true especially for sulfates and for eastern Europe. Indeed, large emissions from combustion plants are accounted for in this region in DJF, but this is not of urban origin, hence this lower relative contribution to the total aerosol burden.

It must be noted, that the radiative impacts of urban emission triggered chemical perturbation presented here are the lower limit we can expect. It has several reasons. 1) due to positive water vapor-cloudiness-precipitation bias in RegCM, the radiative impact will be under-predicted in the model, especially for the aerosols direct effect. 2) both meteorological and chemical lateral boundary conditions (LBC) were kept the same for all the simulations. This causes, especially near the edges of the domain, a suppression of the signal, 3) the radiative impact of secondary organic aerosol (SOA) were not considered. Since fine aerosols

have potentially a significant climate effect, accounting for SOA concentration, optical properties and their radiative forcing is necessary in regional and global climate model studies (Kanakidou et al., 2005).

In Huszar et al. (2014), we examined the climate impact of cities trough the meteorological effects triggered by urban canopy. The impact there exceeded 1° C in JJA and reached about 0.2° C in DJF in absolute values. For both seasons, the absolute climate impact of urban emissions is by more than an order of magnitude smaller. This indicates, at least for central

Europe, that regional climate studies concerning the impact of urban environment on the atmosphere should more focus on this first aspect, i.e. on the impact of urban canopy on the meteorology and climate. However, it must be stressed, that while the climate impact of urban emissions is very small, the impact on air-quality remains significant (Huszar et al., 2016) and has to be considered in future.

*Acknowledgements.* This work has been funded by the Czech Science Foundation (GACR) project No. 13-19733P and by the project

UNCE 204020/2013. We further acknowledge the TNO MEGAPOLI emissions dataset from the EU-FP7 project MEGAPOLI. (http: //megapoli.info).





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





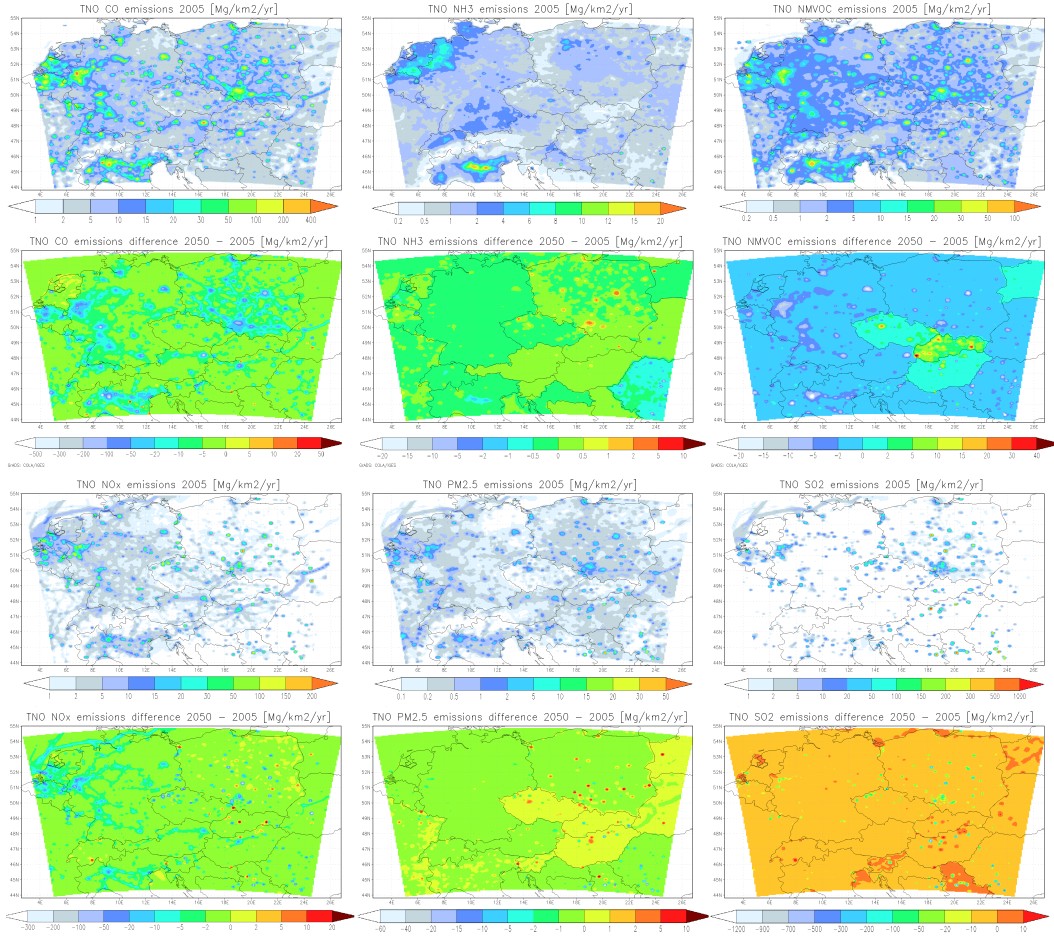

**Figure 1.** Base year (2005) annual emissions (rows 1 and 3) and the projected change between years 2050 and 2005 (rows 2 and 4) for CO, NH$_3$, NMVOC, NOx, PM2.5 and SO$_2$ in $\mathrm{Mg\,yr^{-1}\,km^{-2}}$.



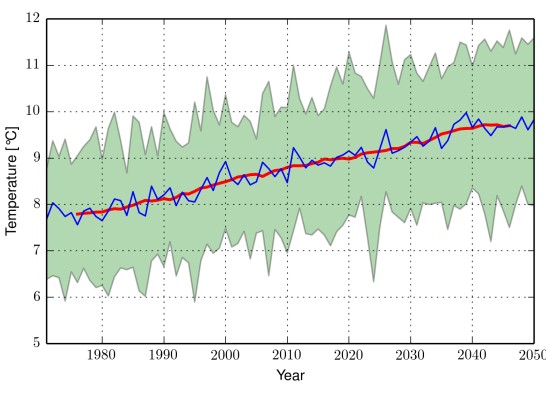

**Figure 2.** The evolution of the annual (blue line) an 10 yr (red line) running mean near surface temperature from 1970 to 2050 averaged over the domain as an ensemble mean of regional climate models from the EURO-CORDEX initiative. The green shade stands for the 10% to 90% ensemble percentiles. For future years, models follow the RCP 4.5 scenario.





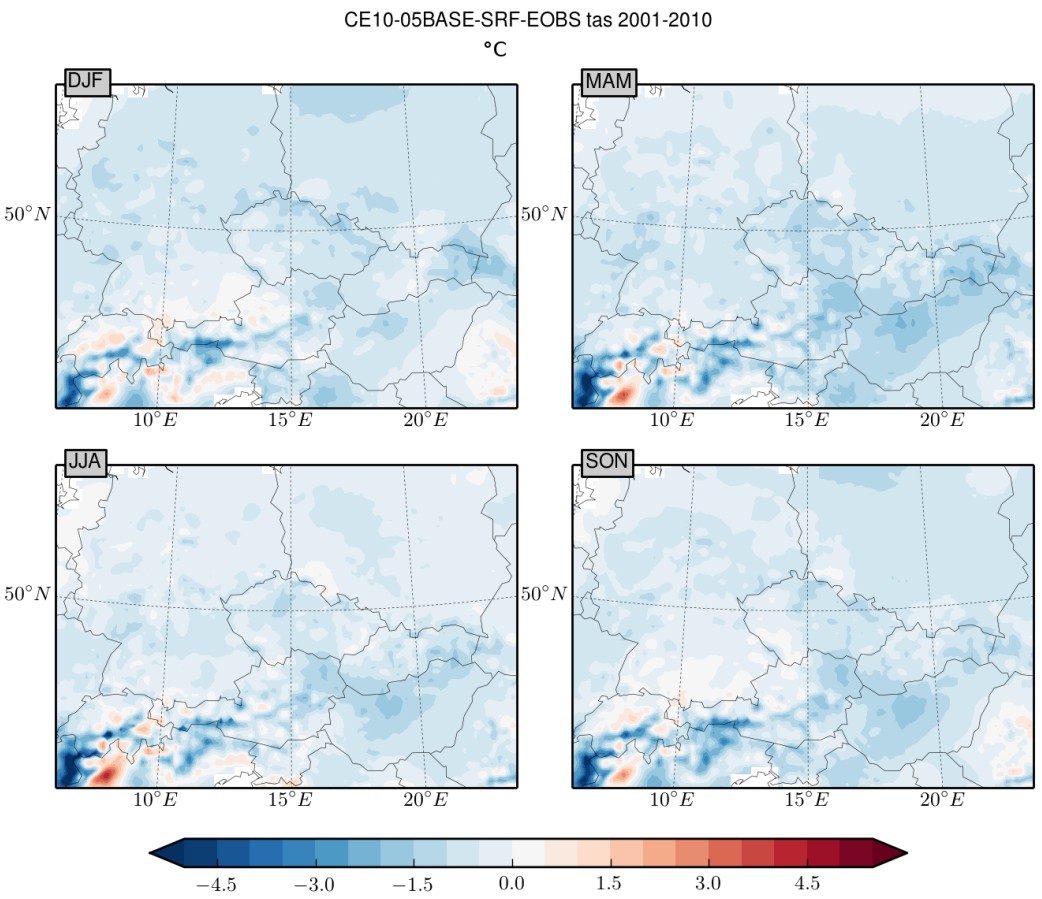

**Figure 3.** Comparison of the modeled seasonal near surface temperatures averaged for the 2001-2010 period with the E-OBS observational data in °C.



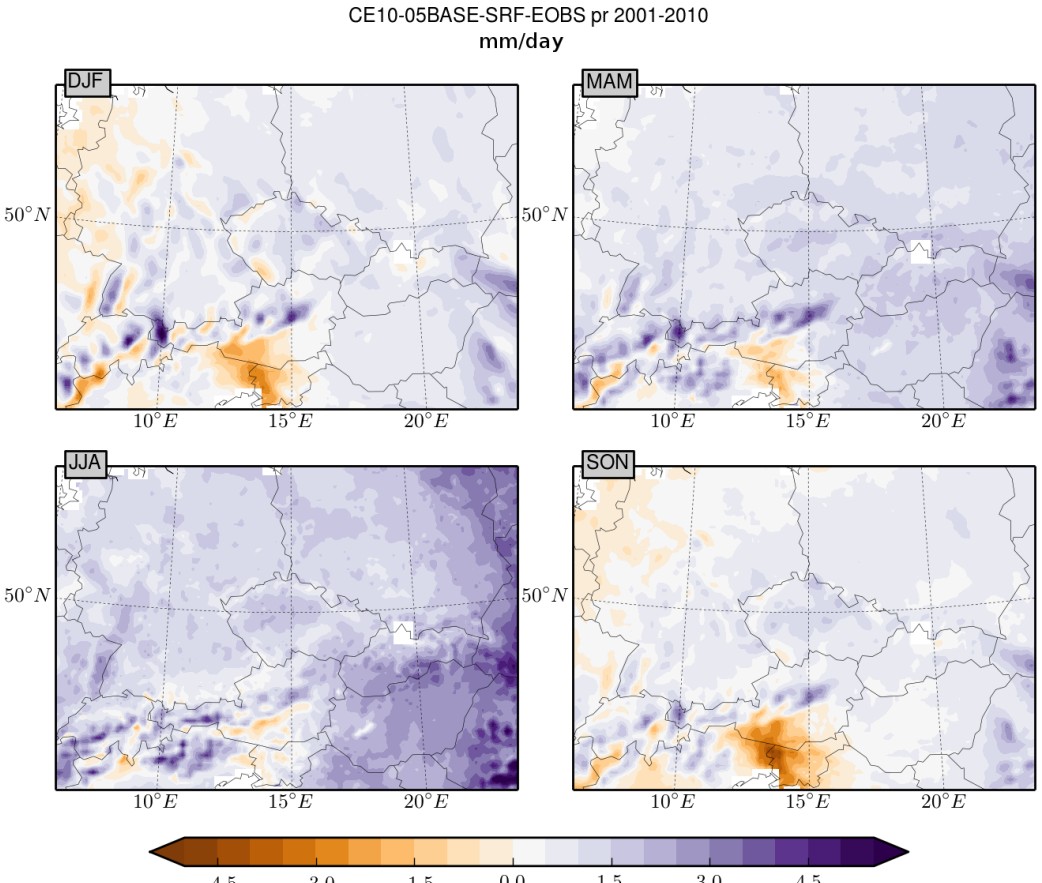

**Figure 4.** Comparison of the modeled seasonal total precipitation (convective and large scale) averaged for the 2001-2010 period with the E-OBS observational data in $\mathrm{mm\,day}^{-1}$.



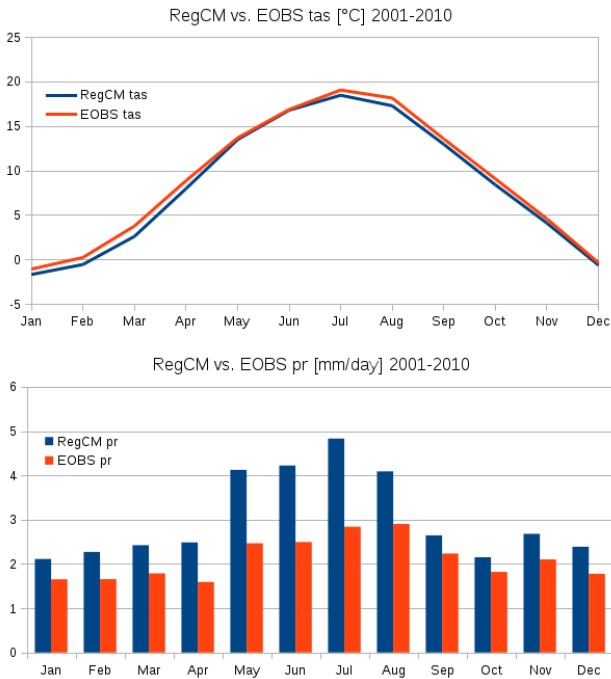

**Figure 5.** The comparison of the modeled domain-averaged annual cycle with observations for near surface temperature (°C) and precipitation [mm day$^{-1}$]

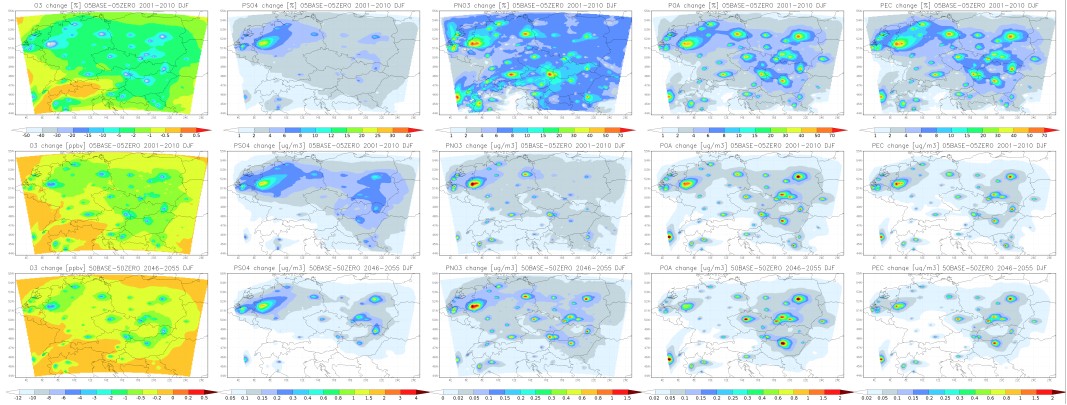

**Figure 6.** The horizontal distribution of the urban emission impact for DJF: upper row the relative impact (for ozone) and relative contribution (for PSO4, PNO3, POA and PEC) in % for the present 2001-2010 period; 2nd row the absolute impact for the present period in ppbv for ozone and μg m$^{-3}$ for aerosols; 3rd row same as the 2nd row but for the future 2046-2055 period.



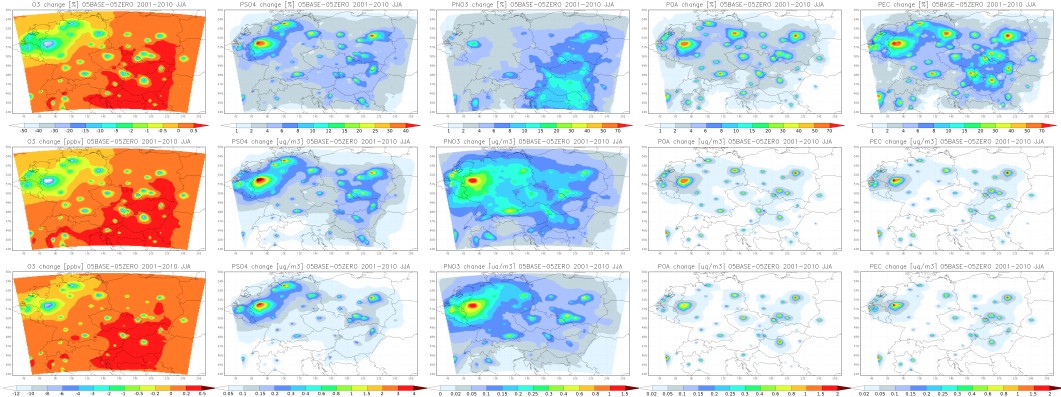

**Figure 7.** The horizontal distribution of the urban emission impact for JJA: upper row the relative impact (for ozone) and relative contribution (for PSO4, PNO3, POA and PEC) in % for the present 2001-2010 period; 2nd row the absolute impact for the present period in ppbv for ozone and µg m$^{-3}$ for aerosols; 3rd row same as the 2nd row but for the future 2046-2055 period.

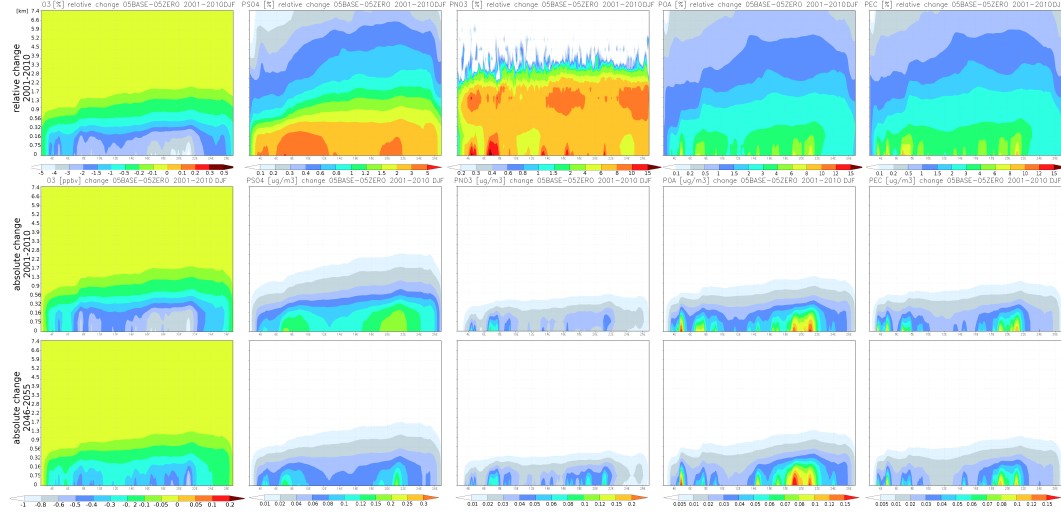

**Figure 8.** The vertical distribution of the urban emission impact averaged over the domain y-axis for winter DJF months: upper row the relative impact (for ozone) and relative contribution (for PSO4, PNO3, POA and PEC) in % for the present 2001-2010 period; 2nd row the absolute impact for the present period in ppbv for ozone and µg m$^{-3}$ for aerosols; 3rd row same as the 2nd row but for the future 2046-2055 period.





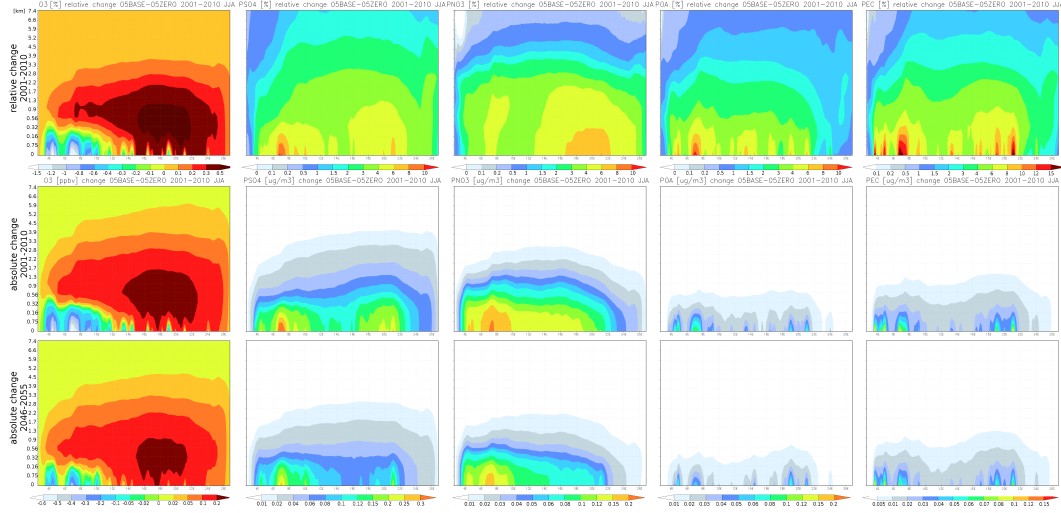

**Figure 9.** The vertical distribution of the urban emission impact averaged over the domain y-axis for summer JJA months: upper row the relative impact (for ozone) and relative contribution (for PSO4, PNO3, POA and PEC) in % for the present 2001-2010 period; 2nd row the absolute impact for the present period in ppbv for ozone and $\mu g\,m^{-3}$ for aerosols; 3rd row same as the 2nd row but for the future 2046-2055 period.

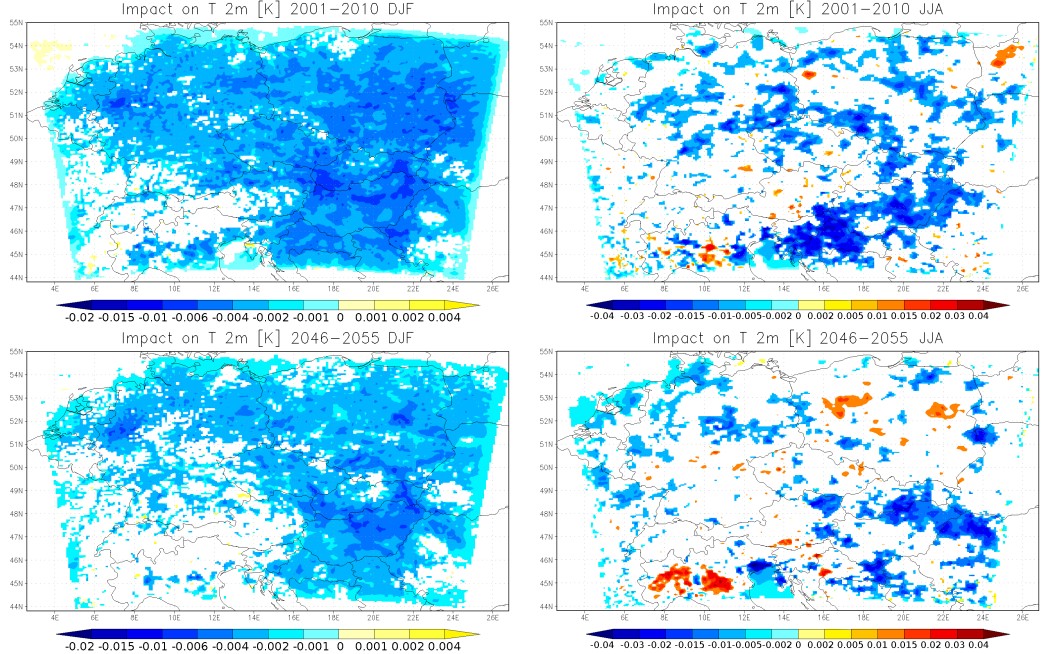

**Figure 10.** Urban emissions impact on near surface temperature in K for DJF (left) and JJA (right) for the 2001-2010 (top) and 2046-2055 (bottom) periods calculated from ensemble averages. Shaded areas represent statistically significant differences on the 98% level.





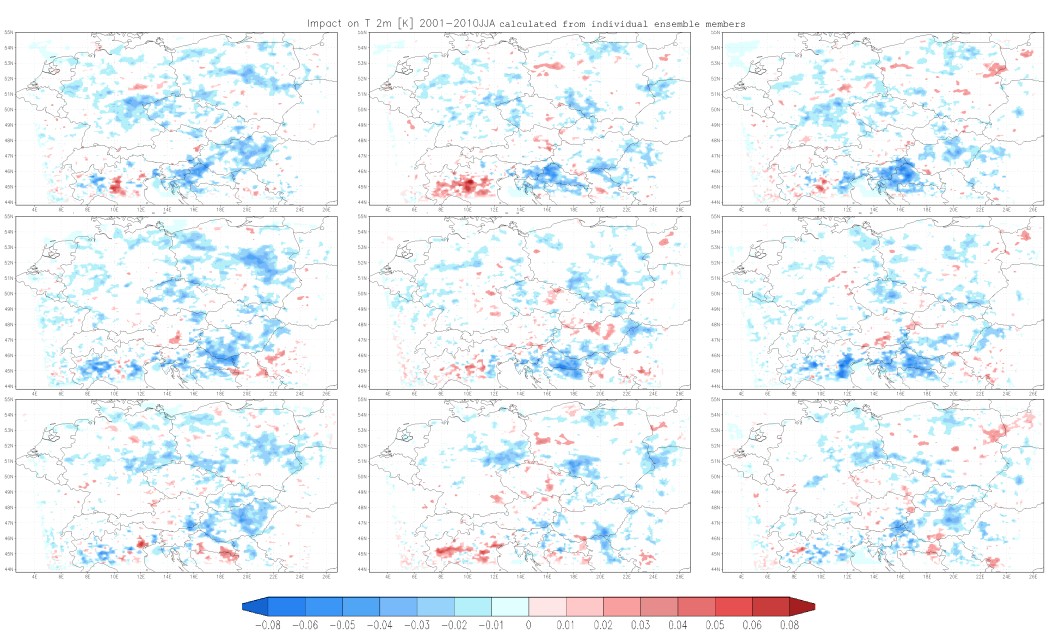

**Figure 11.** Urban emissions impact on near surface temperature in K for JJA (2001-2010) calculated from individual ensemble members. Shaded areas represent statistically significant differences on the 98% level.





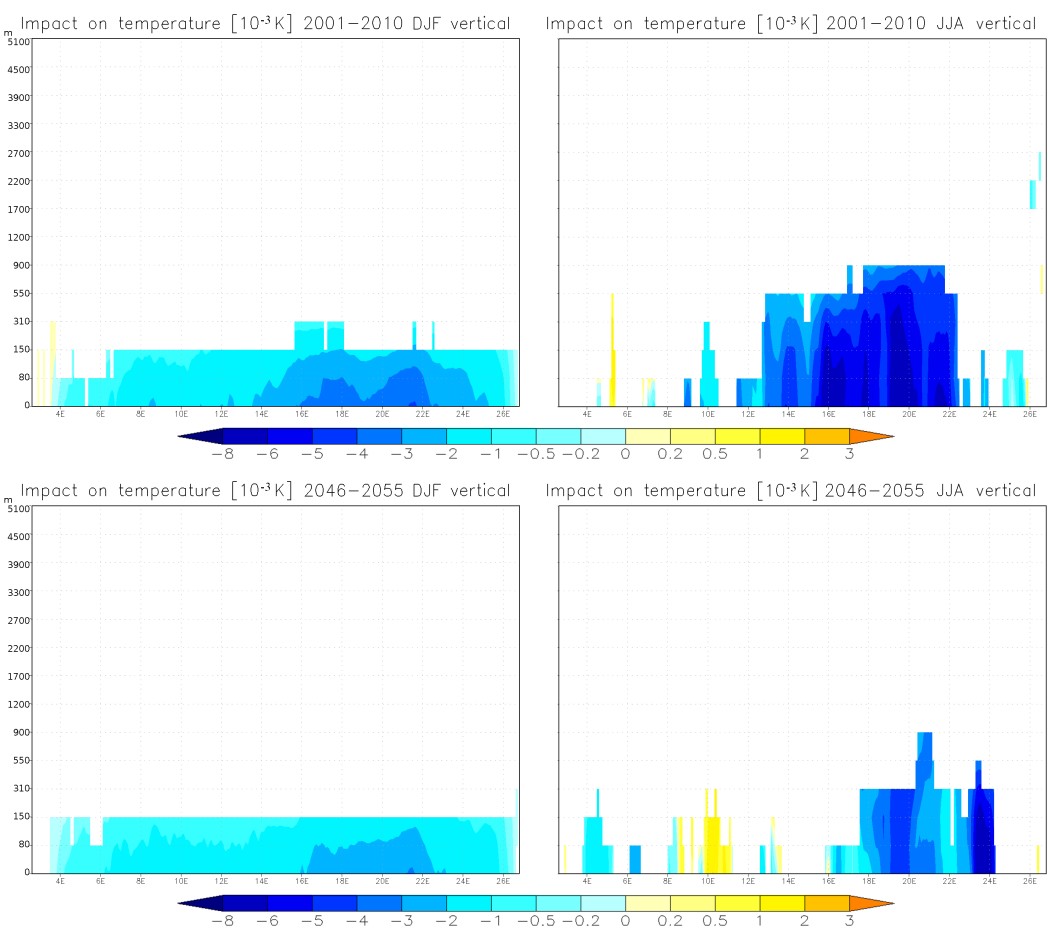

**Figure 12.** The y-axis averaged vertical distribution of the urban emissions impact on temperature in K for DJF (left) and JJA (right) for the 2001-2010 (top) and 2046-2055 (bottom) periods calculated from ensemble averages. Shaded areas represent statistically significant differences on the 98% level.





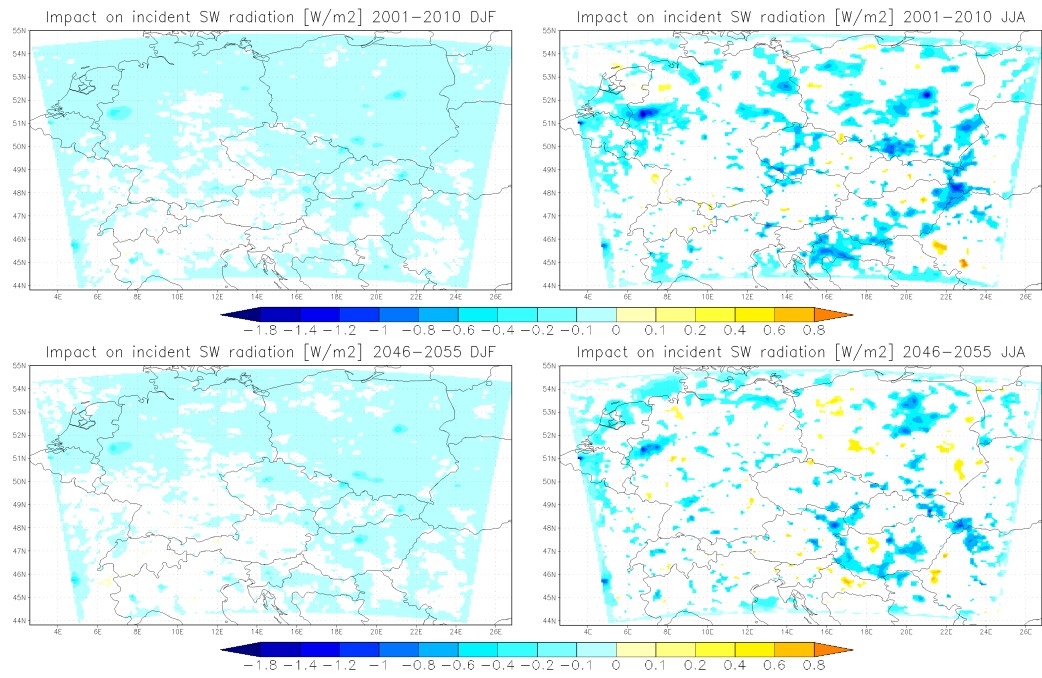

**Figure 13.** Urban emissions impact on the solar incident radiation at the surface in $\mathrm{Wm}^{-2}$ for DJF (left) and JJA (right) for the 2001-2010 (top) and 2046-2055 (bottom) periods calculated from ensemble averages. Shaded areas represent statistically significant differences on the 98% level.





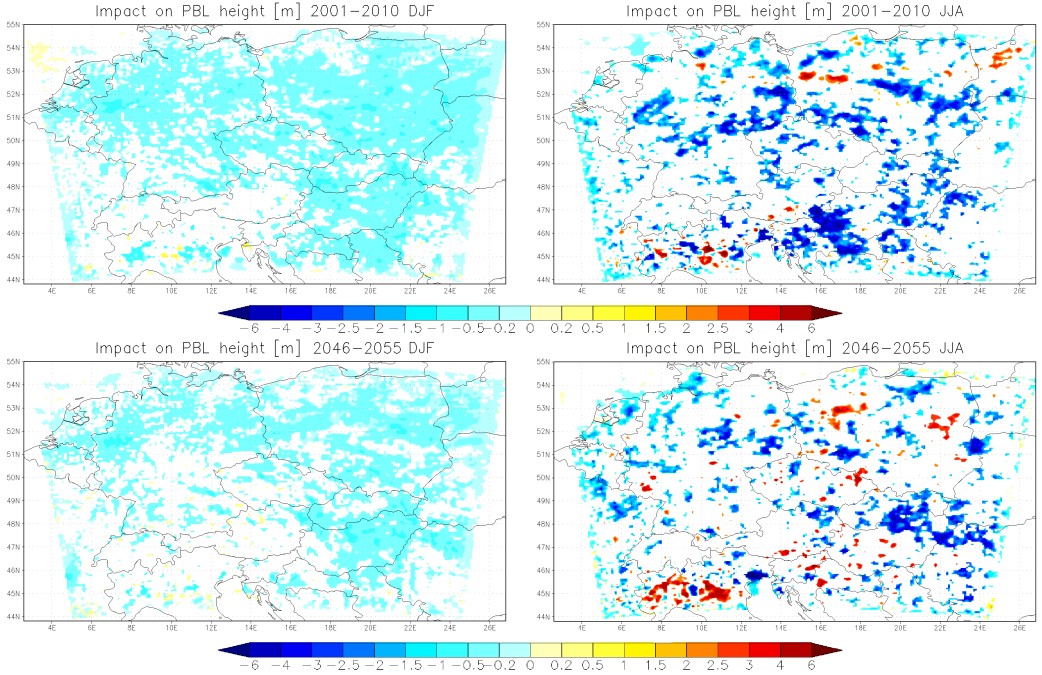

**Figure 14.** Urban emissions impact on the planetary boundary layer height at in m for DJF (left) and JJA (right) for the 2001-2010 (top) and 2046-2055 (bottom) periods calculated from ensemble averages. Shaded areas represent statistically significant differences on the 98% level.

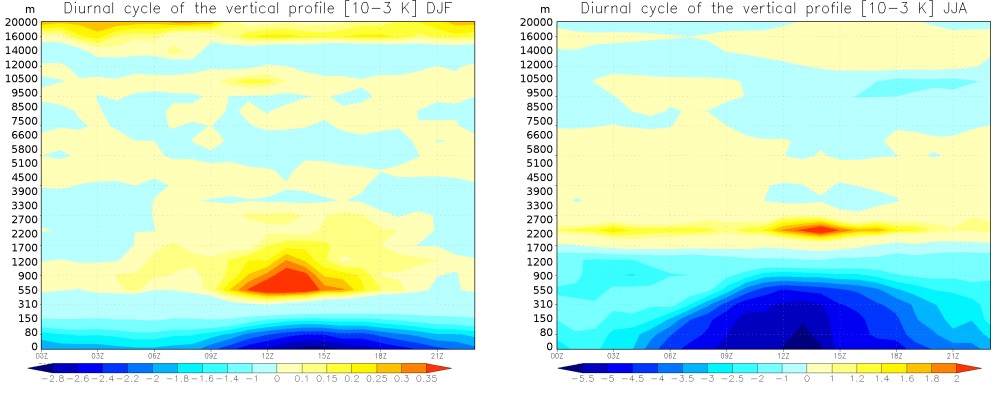

**Figure 15.** Urban emissions impact on temperature: the diurnal cycle of the horizontally averaged vertical profile in $10^{-3}$K for DJF and JJA, present day period (2001-2010).




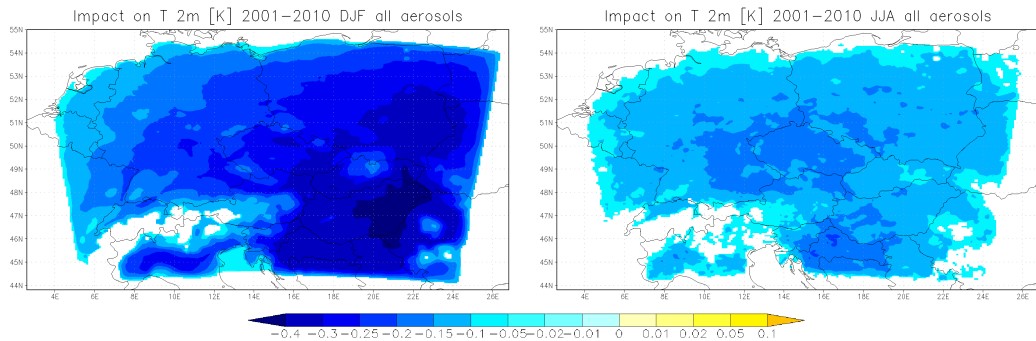

**Figure 16.** Impact of aerosols (PSO4, PNO3, POA and PEC) on near surface temperature in K for DJF (left) and JJA (right) for the 2001-2010 period. Shaded areas represent statistically significant differences on the 98% level.