# Peer review of "The regional impact of urban emissions on climate over central Europe: present and future emission perspective"

_Atmospheric Chemistry and Physics, 2016_

## Referee Comment (RC1) · Anonymous Referee #3 · 14 Jul 2016

The submitted manuscript provides an assessment of regional impact of urban emissions on climate over central Europe for the present and future climate based on the regional climate model RegCM4.2 coupled to the chemistry transport model CAMx. I would suggest acceptance of the manuscript for publication after taking into a number of comments that follow.

Comments Page 5, Sections 2.2: As far as I understand when the authors refer to the experiments for the future period 2046-2055 practically they refer to experiments forced by the ERA-interim meteorology of the decade 2001-2010 with chemical ICBC of the decade 2001-2010 but with anthropogenic emissions of 2050. I think although the authors mention this, it is still somehow misleading the notation for a future simulation over the period 2046-2055. Maybe the authors could make this point even more clear within the manuscript. Page 6, Section 2.4: I think that the authors should provide more details for the individual ensemble members. Page 7, Section 3.1: The authors show differences in Figures 3 and 4 between E-obs gridded data with a resolution of roughly 25 km and RegCM data with 10 km resolution. This can be done either with an upscale interpolation from RegCM towards E-OBS or with a downscale interpolation from E-OBS to RegCM. What was the interpolation procedure that the authors followed? Furthermore the authors should provide information for the used E-OBS data (e.g. version, resolution, reference). Page 7, Section 3.2: Overall it seems from Figure 7 that the urban emissions lead to decrease of ozone over an extended area in Germany as well as at sub-urban and rural areas around the big cities due to NO titration even though it is the summer period. Taking into consideration that there is only slight ozone increase (of up to 0.5 ppbv) for the rest of the domain it could possibly postulated that there is an average ozone decrease for the whole domain due to urban emissions which is somehow not expected for summertime. Maybe it would be insightful if the authors try also to use only the daytime O3 data in order to reduce the effect of nighttime ozone removal (due to NO titration) and discuss this issue. Furthermore, is this slight ozone increase statistically significant? Page 7, Section 3.2: The authors state that the saturated NOx conditions cause ozone titration for the lower model levels. Do the authors mean "with NOx saturated conditions" that ozone production is in the VOC-limited regime or simply refer to the first order ozone removal by NO titration? This point needs clarification. Normally with NOx titration we mean the process of O3 removal through direction reaction with NO which takes place during nighttime and in the vicinity of large NO emission sources. The saturated NOX conditions (or VOC sensitive conditions) is a different issue. The split between NOx-saturated or NOx-sensitive regimes is driven by the chemistry of odd hydrogen radicals with HNO3 being the dominant sink in the first case and peroxides the dominant sink in the second case. Maybe the authors could also refer to the photochemical regimes in their simulations for winter and summer using VOC/NOx or H2O2/NOy ratios (see also the study of Beekmann

and Vautard, ACP, 2010). Page 12, Section 4: It is stated that the model results encounter large wet biases over mountainous areas. This is connected to the convective scheme as high-resolution simulations with the default Grell-FC scheme tend to significantly over estimate precipitation for the mountainous areas as pointed in previous studies with RegCM (Torma et al. 2011; Zanis et al., 2015). Page 12, Section 4: The authors mention that in winter weather is characterized with more stable conditions and reduced variability. In what sense? Do they mean in terms of static stability? Otherwise this statement is wrong as in mid-latitudes, winter is characterized with higher synoptic variability and stronger baroclinic instability. Page 12, Section 4: The authors claim that the maximum cooling is shifted toward later hours due to delayed propagation of aerosol signal through the boundary layer. Is this a speculation or it can be justified from the results of this work or from results of previously published work?

---

## Referee Comment (RC2) · Anonymous Referee #1 · 25 Jul 2016

The regional impact of urban emissions on climate over central Europe: present and future emission perspective

ACPD: doi:10.5194/acp-2016-425, 2016

General comment:

Huszar et al., (2016) makes use of the coupled regional climate model RegCM4.2 and chemistry transport model CAMx with two-way interactions to study the regional impact of urban emission of Central Europe cities on climate for two periods, (2001-2010) and (2046-2055). Numerous simulations are done to evaluate the impacts of ozone and various aerosol species. Authors find that non-CO2 emissions are not the main
players to impact regional climate, while the urban canopy meteorology plays a major role. Despite the authors find negative bias in surface temperature and positive bias in precipitation, the coupling between regional climate model and chemical transport model is a useful and valuable tool and the modeling approach presented is scientifically reasonable. The paper is in general a systematically organized one. I believe the study could benefit the research community after clarifying or providing scientific evidence regarding the items listed below.

Specific comments:

1. P.5, For the future emission used in the study, it is said "A moderate climate policy is assumed . . ." on p.5 (line27), how would the findings change if mitigation strategies in moderate climate policy fails, or not achieving the emission target? Could the authors provide some context about this? Especially, authors remarked in the conclusion (p.13 beginning from line 10), that the radiative impacts of urban emission presented here are at lower limit. It would be essential and interesting to compare the radiative impacts of chemical perturbation under a less-successful mitigation climate policy and to the one presented here for moderate climate policy.

2. P.7, session 3.1 model validation, suggest to write a brief summary about the validation of chemical species here to facilitate the readers' understanding.

3. P.7 and P.10, model biases in surface temperature and precipitation would greatly affect the aerosol loadings within the boundary layer through, for example, wet scavenging and turbulent eddy transport of aerosols. Moreover, the vertical distribution of the aerosol species and ozone would also be affected by surface temperature and precipitation. Although the authors give reasons for the model biases (P.10-P.11 session 4), the author should further elaborate and provide support about the findings in this study is robust or at least not significantly affected by such model biases. Noting that authors already partly addressed (P.13 line 11-12) that the radiative impacts will be under-predicted, it would be good to give some quantitative measure of the underprediction or some sensitivity test results to strengthen the conclusion.

4. P.11 line 30-31, it is about the long-wave heating rate and temperature of air and the surface. It would be more convincing to provide numerical results and/or figures about the vertical heating rate profiles obtained from different simulations. Especially because the model results are known to have negative surface temperature bias.

5. P.12 line 31, suggest to provide results of atmospheric stability obtained from different simulations to further support the argument "...delayed propagation of aerosol signal...".

6. P.13 line 3, please clarify how to reach the number "10% of the total cooling" or relate to other session in the paper. It is not clear to readers how author reach this "10%".

7. P.13 line 22, "...climate impact of urban emissions is very small..." but P.13 line 10, "... lower limit", then it is not legitimate to say "climate impact of urban emission is very small,..." Please rephrase appropriately.

Technical comments:

1. Abstract (line 17), suggest removing the minus sign in "... -1 m and -6 m decreases ...", the minus signs cause confusion. The word "decrease" is sufficient to mean reduction.

2. Abstract (line 22), what do you mean by "... urban surface trough urban canopy meteorological effects"? Do you mean "trough" or "through"?

3. P.2 (line 20-21), suggest rephrasing the sentence, it is too long to be understood.

4. P.3 (line 35), Do you mean "Our region of focus..."?

5. P.11 (line 21), "tinner" should be "thinner"?

---

## Author Comment (AC1) · 24 Aug 2016

Author's response on Anonymous Referee 3 comments:

We would like to thank to Anonymous Referee 3 all the comments, suggestions and corrections in his review of our manuscript. We addressed all and our point-by-point responses including the modifications in the manuscript follow:

*Referee's Comment: Page 5, Sections 2.2: As far as I understand when the authors refer to the experiments for the future period 2046-2055 practically they refer to experiments forced by the ERA-interim meteorology of the decade 2001-2010 with*

[Figure]

*chemical ICBC of the decade 2001-2010 but with anthropogenic emissions of 2050. I think although the authors mention this, it is still somehow misleading the notation for a future simulation over the period 2046-2055. Maybe the authors could make this point even more clear within the manuscript.*

Author's response: We agree that this point is not made enough clear in the manuscript and the reader can be easily mistaken to think that meteorological changes are considered in the future as well (apart from the emission changes of short lived gases/aerosols). So we stressed it in the abstract as well as in the part of the manuscript with the description of the experimental set-up

*Referee's Comment: Page 6, Section 2.4: I think that the authors should provide more details for the individual ensemble members.*

Author's response: We achieved the perturbed runs by changing the RegCM integration step. The base (default) ensemble member was integrated at dt = 30s timestep from the very beginning trough the whole simulation. The further two ensemble members were created by running the model with 1) dt = 15s and 2) dt = 10s timestep for the first few days, and than turning to default 30s timestep for the rest of the simulation. With this we achieved a physically consistent but numerically perturbed ensemble.

We made this clear in the revised manuscript.

*Referee's Comment: Page 7, Section 3.1: The authors show differences in Figures 3 and 4 between E-obs gridded data with a resolution of roughly 25 km and RegCM data with 10 km resolution. This can be done either with an upscale interpolation from RegCM towards E-OBS or with a downscale interpolation from E-OBS to RegCM. What was the interpolation procedure that the authors followed? Furthermore the authors should provide information for the used E-OBS data (e.g. version, resolution, reference).*

Author's response: The E-OBS (version 12.0) gridded observational data was used in the validation (described by Besselaar et al. 2011). Data is available on a 0.25 and 0.5 degree regular lat-lon grid (roughly 15 km x 20 km at the modeled latitudes), which

were interpolated to the model grid (10 km x 10 km).

We included in the revised manuscript a detailed description of the observational dataset used and the interpolation process performed.

*Referee's Comment: Page 7, Section 3.2: Overall it seems from Figure 7 that the urban emissions lead to decrease of ozone over an extended area in Germany as well as at sub-urban and rural areas around the big cities due to NO titration even though it is the summer period. Taking into consideration that there is only slight ozone increase (of up to 0.5 ppbv) for the rest of the domain it could possibly postulated that there is an average ozone decrease for the whole domain due to urban emissions which is somehow not expected for summertime. Maybe it would be insightful if the authors try also to use only the daytime O3 data in order to reduce the effect of nighttime ozone removal (due to NO titration) and discuss this issue. Furthermore, is this slight ozone increase statistically significant?*

Author's response: We found that the main driver for ozone decrease is the first order removal by NO, i.e. NO-titration. In Huszar et al.(2016) we made a sensitivity simulation were NOx emissions were reduced by 20%. This resulted in a strong ozone increase in urban areas, were strong ozone titration occurred. As a domain average, ozone slightly decrease over the first two model layers (i.e. up to about 150 m), however, over higher levels, the domain averaged ozone change is positive. We checked the impact on day-time and night-time average ozone in JJA and found that the daytime changes are although similar in pattern to the "all-day" average, the ozone production far from urban centres is however somewhat stronger (up to 1 ppbv). Regarding the significance, performing t-test showed that the changes are basically significant everywhere (98% level of significance), except those areas where the ozone impact is nearly zero – i.e. the transitional areas between areas where ozone is removed and those with ozone production (see the attached Fig. 1 below).

In the revised manuscript, we added some notes considering this issue, however we do not want to add more figures as the chemical behavior due to urban emissions was already discussed in Huszar et al.(2016) and this paper is intended to focus on the

meteorological/climate effects.

*Referee's Comment: Page 7, Section 3.2: The authors state that the saturated NOx conditions cause ozone titration for the lower model levels. Do the authors mean "with NOx saturated conditions" that ozone production is in the VOC-limited regime or simply refer to the first order ozone removal by NO titration? This point needs clarification. Normally with NOx titration we mean the process of O3 removal through direction reaction with NO which takes place during nighttime and in the vicinity of large NO emission sources. The saturated NOX conditions (or VOC sensitive conditions) is a different issue. The split between NOx-saturated or Nox-sensitive regimes is driven by the chemistry of odd hydrogen radicals with HNO3 being the dominant sink in the first case and peroxides the dominant sink in the second case. Maybe the authors could also refer to the photochemical regimes in their simulations for winter and summer using VOC/NOx or H2O2/NOy ratios (see also the study of Beekmann and Vautard, ACP, 2010).*

Author's response: We agree that the term saturated NOx conditions along with titration is not correct, as here we really meant first order ozone removal by NO. Indeed, in Huszar et al. (2015) we showed that simultaneous reduction of NO+NO2 emissions leads to ozone increase over urban areas due to less available NO entering the O3+NO reaction. Only further from cities was found the NOx emission perturbation without any effect, which indicates NOx-saturated conditions. But again, in the manuscript, we meant NO titration as the main cause for ozone decrease.

We also agree that it would be interesting to see the split between NOx-saturated or NOx-sensitive regimes by evaluating the suggested ratios, however this would go beyond the scope of the paper so we decided to only make a clarification of the ozone changes from the point of view of interactions with NOx emissions.

*Referee's Comment: Page 12, Section 4: It is stated that the model results encounter large wet biases over mountainous areas. This is connected to the convective scheme as high-resolution simulations with the default Grell-FC scheme tend to significantly over estimate precipitation for the mountainous areas as pointed in previous studies*

*with RegCM (Torma et al. 2011; Zanis et al., 2015).*

Author's response: We agree that the use of Grell scheme with the Kain-Fritch closure could be another source of precipitation overestimation over mountainous areas.

We included this into the revised manuscript and the suggested references as well.

*Referee's Comment: Page 12, Section 4: The authors mention that in winter weather is characterized with more stable conditions and reduced variability. In what sense? Do they mean in terms of static stability? Otherwise this statement is wrong as in mid-latitudes, winter is characterized with higher synoptic variability and stronger baroclinic instability.*

Author's response: Here, we meant reduced instability or in other words, increased stability in winter. this greatly influences the vertical mixing and thus the vertical extent of the urban impact.

We made this point clear in the revised manuscript.

*Referee's Comment: Page 12, Section 4: The authors claim that the maximum cooling is shifted toward later hours due to delayed propagation of aerosol signal through the boundary layer. Is this a speculation or it can be justified from the results of this work or from results of previously published work?*

Author's response: This was based on a rather speculation, however we reconsidered the causes for this shift and found that the main reason for this is probably the shifted maximum aerosols concentration towards later hours in the afternoon in DJF compared to JJA.

To support this, we added a figure (Fig. 17) (and an accompanying discussion) to the manuscript's Discussion section showing the diurnal cycle of the vertical distribution of the urban impact on aerosols.

References

van den Besselaar, E.J.M., Haylock, M. R., van der Schrier, G., and Klein Tank, A. M. G.: A European Daily High-resolution Observational Gridded Data set of Sea

[Figure]

Level Pressure, J. Geophys. Res., 116, D11110, doi:10.1029/2010JD015468, 2011.

Huszar, P., Belda, M., and Halenka, T.: On the long-term impact of emissions from central European cities on regional air quality, Atmos. Chem. Phys., 16, 1331–1352, doi:10.5194/acp-16-1331-2016, 2016.

[Figure]

**O3 change [ppbv] 05BASE-05ZERO 2001-2010 JJA day 98%**

**Fig. 1.** The urban emission induced day-time ozone changes. Shaded areas mean singificant changes on the 98\% level using t-test.

---

## Author Comment (AC2) · 24 Aug 2016

Author's response on Anonymous Referee 1 comments:

We would like to thank to Anonymous Referee 1 for his comments, suggestions and corrections. We addressed all and our point-by-point responses including the modifications in the manuscript follow:

*Referee's Comment: 1. P.5, For the future emission used in the study, it is said "A moderate climate policy is assumed on p.5 (line27), how would the findings change if mitigation strategies in moderate climate policy fails, or not achieving the emission*

[Figure]

*target? Could the authors provide some context about this? Especially, authors remarked in the conclusion (p.13 beginning from line 10), that the radiative impacts of urban emission presented here are at lower limit. It would be essential and interesting to compare the radiative impacts of chemical perturbation under a less-successful mitigation climate policy and to the one presented here for moderate climate policy.*

Author's response: The intention of the study was to 1) evaluate the urban emission's impact for present day emissions and 2) the same impact in future assuming a future scenario of emissions. As the future emissions are lower (except certain activity sectors) in general in this scenario, the impact itself was (as expected) quantified to be also lower a bit. One can conclude that without achieving the emission reduction goals for the middle of the century (for which we had the emission scenario), which would mean that emissions would stay at the present level, the radiative impact of urban emission would not change.

We added a small paragraph to the description of the emissions/scenarios to clarify this point.

Referee's Comment: P.7, session 3.1 model validation, suggest to write a brief summary about the validation of chemical species here to facilitate the readers' understanding.

Author's response: We agree that a brief summary of the most important points regarding the chemical validation (presented in Huszar et al. 2016) is necessary to include in order to facilitate the reader's understanding of the model's overall performance.

We therefor added a paragraph to address the most important findings during the chemical validation.

*Referee's Comment: P.7 and P.10, model biases in surface temperature and precipitation would greatly affect the aerosol loadings within the boundary layer through, for example, wet scavenging and turbulent eddy transport of aerosols. Moreover, the vertical distribution of the aerosol species and ozone would also be affected by surface temperature and precipitation. Although the authors give reasons for the model biases*

*(P.10-P.11 session 4), the author should further elaborate and provide support about the findings in this study is robust or at least not significantly affected by such model biases. Noting that authors already partly addressed (P.13 line 11-12) that the radiative impacts will be under-predicted, it would be good to give some quantitative measure of the underprediction or some sensitivity test results to strengthen the conclusion.*

Author's response: We completely agree that the manuscript misses a discussion of how the model biases affect the results. In this context the precipitation bias is the most striking and therefore we made two additional sensitivity runs for a shorter, 2001-2005 period: one experiment for the base (05BASE) case and one for the zero (05ZERO) case. In these runs the precipitation fields from RegCM were reduced by 30

We added a paragraph in the Conclusion section to address this issue. We also included a figure (Fig.18) as an example for the increase of aerosol concentration (for nitrates – for other aerosols the response is similar, and for ozone it is almost negligible).

*Referee's Comment: P.11 line 30-31, it is about the long-wave heating rate and temperature of air and the surface. It would be more convincing to provide numerical results and/or figures about the vertical heating rate profiles obtained from different simulations. Especially because the model results are known to have negative surface temperature bias.*

Author's response: The corresponding paragraph tries to explain the reader that the main driver for the simulated radiative/temperature changes is the aerosol enhancement, i.e. less shortwave radiation reaching the surface. However, one must consider also the impact of the ozone changes which encompass both reduction near the surface and large sources, and production further from them and above higher elevations. Here, we refer to previous studies that concluded that ozone LW cooling rates in the lower troposphere (where ozone changes are significant in our study) are very small due to similar temperatures with the surface (Petty, 2006; Liou, 2002). This means that any perturbation of ozone concentrations near the surface have a small

longwave radiative impact - unlike for aerosols, that in our study interact only with short wave radiation (in the used version of RegCM4 only large dust particles interact with LW radiation).

We agree that it would be conclusive to evaluate the short-wave heating and long-wave cooling rates for aerosols and ozone separately, but this is impossible within the modeling framework at this stage (i.e. one cannot output individual rates for each species), so we would like to rely purely on the cited literature about ozone heating/cooling rates and that of aerosols.

*Referee's Comment: P.12 line 31, suggest to provide results of atmospheric stability obtained from different simulations to further support the argument "delayed propagation of aerosol signal..."*

Author's response: We re-evaluated the causes of the delayed surface response of the winter urban aerosol enhancement by analyzing the diurnal cycle of the aerosol vertical distribution. We found that there is a shift of maximum aerosol load towards later hours (due to the emission distribution during the day). Because of the shift of the maximum values there is a slight shift also in the radiative impacts.

This has been clarified in the manuscript and a figure has been added for DJF and JJA showing the diurnal cycle of the domain averaged vertical profile of urban induced aerosol load (those aerosol types that we considered in the radiative calculation)

*Referee's Comment: 6. P.13 line 3, please clarify how to reach the number "10% of the total cooling" or relate to other session in the paper. It is not clear to readers how author reach this "10%".*

Author's response: In the revised manuscript, we made this clear: the cooling by urban aerosol (up to about 0.05K) represents approximately 10% of the cooling caused by all (i.e. not only urban induced) aerosol (up to about 0.5 K).

*Referee's Comment: P.13 line 22, "...climate impact of urban emissions is very small..." but P.13 line 10,"...lower limit", then it is not legitimate to say "climate impact of urban emission is very small". Please rephrase appropriately.*

Author's response: We agree that such a formulation is not legitimate, as in theory, if

it is only a lower limit, the actual effect could be higher, becoming no longer "small". We reformulated the whole paragraph to stress that even without these shortcomings, the urban impact on radiation/climate would not change (increase) significantly as 1)the most important urban aerosols were considered, showing that the disregarded SOA comprises about 10% of the aerosols considered in radiation calculation 2) the effect of fixed boundary conditions is significant in the outer part of the domain only and becomes minimal in the inner part 3) the sensitivity runs showed that without the modeled precipitation bias there would be even larger temperature decrease, but this change is of one order less than the overall cooling. We made these points clear in the Discussion part of the revised manuscript.

Technical corrections

Author's response: We implemented in the revised manuscript all the technical corrections suggested by the referee.

References:

Huszar, P., Belda, M., and Halenka, T.: On the long-term impact of emissions from central European cities on regional air quality, Atmos. Chem. Phys., 16, 1331–1352, doi:10.5194/acp-16-1331-2016, 2016.

Liou, K. N.: An Introduction to Atmospheric Radiation, Academic Press, San Diego, USA, 2002.

Petty, G. W.: A First Course in Atmospheric Radiation (2nd. Ed.), Sundog Publishing, Madison, Wisconsin, 2006.